# Cause-Specific Mortality Fraction (CSMF) of adult mortality in Butajira, South Central Ethiopia

**Hailelule Aleme** ***, Wubegzier Mekonnen, Alemayehu Worku**

School of Public Health, College of Health Sciences, Addis Ababa University, Addis Ababa, Ethiopia

* hailelule.aleme@aau.edu.et

**Data Availability Statement:** The minimum anonymized or de-identified data to generate the results are within the paper and Supporting Information files.

## Abstract

Cause- and context-specific mortality data are imperative to understand the extent of health problems in low-income settings, where national death registration and cause of death identification systems are at a rudimentary stage. Aiming to estimate cause-specific mortality fractions, adult (15+ years) deaths between January 2008 and April 2020 were extracted from the Butajira health and demographic surveillance system electronic database. The physician review and a computerized algorithm, InterVA (Interpreting Verbal Autopsy), methods were used to assign the likely causes of death from January 2008 to April 2017 (the first) and May 2017 to April 2020 (the second) phase of the surveillance period, respectively. Initially, adult mortality rates per 1000py across sex and age were summarized. A total of 1,625 deaths were captured in 280, 461 person-years, with an overall mortality rate of 5.8 (95%CI: 5.5, 6.0) per 1000py. Principally, mortality fractions for each specific cause of death were estimated, and for 1,571 deaths, specific causes were determined. During the first phase, the leading cause of death was tuberculosis (13.6%), followed by hypertension (6.6%) and chronic liver disease (5.9%). During the second phase, digestive neoplasms (17.3%), tuberculosis (12.1%), and stroke (9.4%) were the leading causes of death, respectively. Moreover, tuberculosis was higher among persons aged 50+ (15.0%), males (13.8%), and in rural areas (14.1%) during the first phase. Hypertensive diseases were higher among females (7.9%) and in urbanities. In the second phase, digestive neoplasms were higher in the age group of 50–64 years (25.4%) and females (19.0%), and stroke was higher in older adults (65+) (10%) and marginally higher among males (9.7%). Our results showed that tuberculosis and digestive neoplasms were the most common causes of death. Hence, prevention, early detection, and management of cases at all levels of the existing healthcare system should be prioritized to avert premature mortality.

## Introduction

Empirical information on the population level mortality and cause of death (CoD) data is imperative to understand the burden of health problems [1]. It provides crucial policy inputs

**Funding:** The authors received no specific funding for this work.

**Competing interests:** The authors have declared that no competing interests exist.

for a well-informed decision-making process and public health measures to prevent premature deaths [2]. High–income countries generally have robust death registration and CoD identification systems [3]. However, these approaches are a rudimentary stage of development in most low-income countries where mortality is highest [4, 5], and many deaths are unregistered and unnoticed [4]. The death registration levels in most African countries, including Ethiopia, are low, and the recording and compilation of CoD are very limited [6]. In low resource-limiting settings, population-based CoD data for deaths happening in the demographic surveillance area (DSA) can be obtained through verbal autopsy (VA) [5], primarily from the Health and Demographic Surveillance System (HDSS) sites [7]. In the HDSS, verbal autopsy (VA) data are routinely obtained from systematically recognized and documented deaths during each household visit. For over two decades and in over 45 Low and Middle-income countries (LMICs), VA has been a well-functioning surveillance method with significant policy influence [8].

VA is a tool for understanding the population-level patterns of cause-specific mortality fractions (CSMFs) in settings where disease burdens are most significant and routine cause-specific mortality data do not exist [9, 10]. Moreover, VA findings are most suitable for determining CSMFs [10, 11]. Although not without limitations, VA reliably estimated CSMFs for diseases of public health importance [9]. Population-level CoD distribution and patterns measured by CSMFs are helpful [12], and there is an increased interest in VA as a tool for measuring population CSMFs in such resource-limiting settings [9, 13]. Recently, concerning the methodological advances in applying VA to determine CoD [14], different VA interpretation methods have been emerging [15]. Conventionally, the underlying CoD has been determined by a physician [16]; however, since 2000, the InterVA4 algorithm has been used for interpreting VA data [16, 17].

Verbal autopsy-based CSMF estimate in Ethiopia indicated 15.9% [18] and 19.7% [19] of deaths from tuberculosis, 8.9% from HIV [19], and 31.9% from hypertension [20]. Further reports also indicated that in Ethiopia, tuberculosis-related mortality declined from 89 per100,000 in 1990 to 24 per100,000 in 2017. However, the country is still among high tuberculosis burden countries [21]. Though AIDS-related deaths reduced from 83,000 in 2000 to 15,600 in 2017, various challenges face the primary prevention practices [22]. The WHO regional report also indicated that hypertension affects 15.9% of the population in Ethiopia, where only 1.5% of it has been controlled, and 2.8% are getting appropriate treatment and care [23]. Recent reports also indicated that the number of new cancer cases in Ethiopia was 77,352, with 51,865 deaths [24]. Moreover, according to the latest WHO report, road traffic deaths in Ethiopia reached 31,564, or 5.60% of total deaths [25].

In Ethiopia, the death registration and CoD identification systems are at an infant stage and highlight the importance of routine VA in determining CoD. It is a cost-effective and interim solution to fill the gaps in death registration and CoD determination activities. Hence, such information needs to be continuously documented, which warrants conducting this study using the Butajira HDSS database.

## Methods

### Study area and setting

Butajira HDSS is located in south-central Ethiopia within the Guraghe zone of the Southern Nations, Nationalities, and People's Region (SNNPR), approximately 130 kilometres south of Addis Ababa, the capital city of Ethiopia. The Butajira HDSS covers households in nine rural and one urban study village/kebele (the smallest administrative unit) selected using the proportional probability to size technique from eighty-two rural and four urban Kebeles. Butajira HDSS employed a baseline census conducted in 1987, which periodically updated the vital and

migratory status changes. Hence for such practices, monthly active household visits (until 1999) and quarterly visit (after 1999) has been conducted. In addition, censuses were conducted every 3–5 years, which was changed to housing, individual and family reconciliation in 2003/2004.

## Data collection and analysis

Village-based data collectors manually record socio-demographic and economic characteristics of individuals during birth, death, marital status, migration, and household formation registration. In addition, a VA has been conducted in all deaths after the culturally accepted mourning period of 40 days. The VA involves a structured interview with the next of kin or a caregiver about signs, symptoms, and events the deceased experienced before death [26]. The CoD identification team has conducted a standardized VA procedure, using the WHO verbal autopsy tool customized to the specific setting to capture signs, symptoms, and events leading to death. All deaths happening in the demographic surveillance area are captured, and the CoD is determined. All adult deaths collected from January 2008 to April 2020 on usual residents in the Butajira HDSS catchment area were considered for this analysis. This study considered the deaths of adults aged 15 years and above. Moreover, the age group 15–49, 50–64, and 65 years and above were labeled as young adults, middle-aged adults, and older adults, respectively, for age disaggregated analysis [26].

Death captured in the first surveillance period, from January 2008 to April 2020, used the physician review method to assign the underlying CoD. On the other hand, for deaths happening in the second surveillance phase from May 2017 to April 2020, the underlying CoD was determined using the computer-coded verbal autopsy model called InterVA. The agreement between the cause of death interpretation methods, the InterVA models, and a physician-certified verbal autopsy was assessed with concordant findings [27]. The two methods used the International Classification of Diseases 10th Edition (ICD-10) as a standard. In the physician review procedure to assign the underlying CoD, two physicians independently reviewed the VA questionnaires and assigned a single probable underlying CoD, considered the final CoD, If the two physicians gave an identical ICD-10 code. If not, a third reviewer is invited to review the questionnaire. If two of these three physicians agreed to assign a similar ICD-10 code, the final CoD was assigned. If not, the CoD is classified as "indeterminate" [28]. The InterVA model, a computer program that applies Bayesian probabilistic modelling, is used to assign a probable underlying CoD. InterVA-4 generates up to three probable causes of death for each case with their assigned statistical likelihoods or indeterminate result after the completed verbal autopsy data are entered into the InterVA-4 algorithm. The sum of the likelihoods of assigned causes has a maximum value of one. If no single cause has a final probability of at least 0.4 (the 0.4 cut-off is considerably higher than the unconditional probability of any cause) or if the sum of their likelihoods was less than one, then the residual component was "indeterminate." The details were discussed elsewhere [27]. The InterVA4 model version 4.02 was used to determine the CoD for the dataset of the second surveillance period. Since Malaria and HIV/AIDS prevalence varies from place to place, they were used as basic epidemiological parameters for the model. Reports indicate that malaria and HIV/AIDS prevalence in Ethiopia is low [29, 30]. Hence for this computer-coded probabilistic expert-driven InterVA model, they predefined as "low."

Adult mortality rates/1000py were estimated across sex and for the predefined age groups. Besides, age, sex, and cause-specific mortality fractions (CSMFs) were computed for the probable underlying CoD and their residual indeterminate fraction for the specific disease classification (S1 Text). For better understanding, the specific CoD determined using the physician-

coded verbal autopsy (PCVA) and the computer-coded verbal autopsy (CCVA) methods were presented together according to their follow-up years. Preliminary data editing and summarizations were done using a Microsoft Excel worksheet, and R- Statistical software package version 3.6.3 was employed for major statistical computation.

## Ethical considerations

This study was done after getting ethical clearance from the Research Ethics Committee (REC) of the School of Public Health and the Institutional Review Board of the College of Health Sciences in Addis Ababa University with protocol number: 047/19/SPH. Besides, permission was also obtained from the Butajira HDSS management committee to receive the dataset.

## Results

### Socio-demographic characteristics of the deceased

During the study period of January 2008 to April 2020, a total of 1,625 adult deaths were captured by the HDSS in 280, 461 person-years, of which 1,571 deaths (96.7%) were assigned a specific underlying CoD while the rest 54 cases (fifty cases from physician review method and four cases from the InterVA algorithm) were labelled as "indeterminate." The median age at death was 65 years (Inter Quartile Range [IQR] = 49, 80 years). A large proportion of deaths, 1,287 (81.9%), were from rural areas, and 284 (18.1%) were from urban areas. The majority of the deaths,1,190 (75.7%), happened in the study sites where people were not able to read and write, 902 (57.4%) were married, and 637 (40.5%) of the deceased were farmers by occupation. Besides, 1,432 (91.1%) deaths happened at home (Table 1).

**Table 1. Socio-demographic characteristics of the deceased, Butajira HDSS, January 2008- April 2020, Ethiopia.**

| | | Age category (n = 1,571) | | | | | |
| --- | --- | --- | --- | --- | --- | --- | --- |
| | | 15–49 | | 50–64 | | 65 + | |
| Variable | Category | N | % | N | % | N | % |
| Sex | Female | 171 | 43.5 | 167 | 52.4 | 452 | 52.6 |
| | Male | 222 | 56.5 | 152 | 47.6 | 407 | 47.4 |
| Marital status | Married | 230 | 58.5 | 216 | 67.7 | 456 | 53.1 |
| | Single | 119 | 30.3 | 14 | 4.4 | 21 | 2.4 |
| | Divorced | 10 | 2.5 | 8 | 2.5 | 25 | 2.9 |
| | Widowed | 34 | 8.7 | 81 | 25.4 | 357 | 47.0 |
| Occupation | Farmer | 126 | 32.1 | 138 | 43.3 | 373 | 43.4 |
| | Housewife | 82 | 20.9 | 107 | 33.5 | 275 | 32.0 |
| | Merchant | 68 | 17.3 | 38 | 11.9 | 99 | 11.5 |
| | Other | 117 | 29.8 | 36 | 11.3 | 112 | 13.0 |
| Education | Illiterate | 176 | 44.8 | 253 | 79.3 | 761 | 88.6 |
| | Primary education | 166 | 42.2 | 53 | 16.6 | 83 | 9.7 |
| | Secondary education | 38 | 9.7 | 6 | 1.9 | 8 | .9.0 |
| | Higher education | 13 | 3.3 | 7 | 2.2 | 7 | .8.0 |
| Residence | Urban | 91 | 23.2 | 50 | 15.7 | 143 | 16.6 |
| | Rural | 302 | 76.8 | 269 | 84.3 | 716 | 83.4 |
| Place of death | Health institution | 33 | 8.4 | 17 | 5.3 | 48 | 5.6 |
| | Home | 339 | 86.3 | 291 | 91.2 | 802 | 93.4 |
| | Other place | 21 | 5.3 | 11 | 3.4 | 9 | 1.0 |

## Crude and disaggregated mortality rates

A total of 1,625 deaths were captured in 280, 461 person-years, with an overall mortality rate of 5.8 (95%CI: 5.5, 6.0) per 1000py. The mortality rate was further disaggregated by age and gender. The mortality rate across the predefined age group indicated 1.7, 15.8, and 48.9 in the age group 15–49, 50–64, and 65 +, respectively. Moreover, comparable mortality rates per 1000py were observed among males (5.9) and females (5.8) (Table 2).

## Cause specific mortality fractions

During the first phase of the surveillance period between January 2008 and April 2017, 1,166 deaths were assigned a specific COD, and CSMFs were computed for each cause. Moreover, specific causes of death were disaggregated by sex, age, and residence. Of the total deaths in the surveillance years, 313, 253, and 600 were determined in the age group 15–49, 50–65, and 65+, respectively. Besides, the cause-specific mortality fractions were disaggregated by age-sex dimensions for the top twenty specific causes during the first phase of the surveillance period (Table 3).

Variations in the patterns of death were demonstrated across and within each age group. During the follow-up period, tuberculosis was the leading CoD and accounted for 13.6% of all deaths, followed by hypertensive diseases (6.6%) and chronic liver disease (5.9%). Deaths from HIV/AIDS, pulmonary tuberculosis, and chronic liver disease in young adults (15–49 years) accounted for 13.4%, 10.2%, and 8.3% and were the first, second, and third leading CoD, respectively. Pulmonary tuberculosis (15.4%), hypertensive diseases (7.5%), and chronic liver disease (7.1%) were the first, second, and third causes of death among middle-aged adults (50–64 years of age). Moreover, pulmonary tuberculosis, acute lower respiratory infections, and hypertensive diseases were the first, second, and third leading causes of death and responsible for 14.5%, 7.8%, and 7.7% of the deaths in older adults (65+).

Fig 1 below reveals the probability of dying from ten top specific causes of death across and within the predefined age category. The CSMF of the overall death showed that across the range of causes of death, a randomly selected case had a significantly higher probability of death from tuberculosis. Hypertensive diseases have also responsible for the higher probability of CoD in the surveillance period (Fig 1).

In the second phase of the follow-up time, from May 2017 to April 2020, 405 deaths were assigned a specific CoD, which contains verbal autopsy titles with the WHO VA CoD code. Similarly, the cause-specific mortality fraction for all adult deaths in the Butajira HDSS/DSA, as shown in Table 4 below, depicts the age-sex disaggregated cause-specific mortality fractions for the top twenty specific causes (Table 4).

In these surveillance years, CSMF was also computed for each specific CoD. A total of 80, 66, and 259 deaths were determined in the age group 15–49, 50–65, and 65+, respectively. Variations in the patterns of death were demonstrated across and within each age group. In the

**Table 2. Adult mortality rate per 1000py, Butajira HDSS, January 2008-April 2020, Ethiopia.**

| Age and sex category (n = 1,625) | | | | |
|---|---|---|---|---|
| Variable | Category | N | Person-year(py) | Mortality rate/1000 py (95% CI) |
| Sex | Male | 809 | 138,787 | 5.9 (5.5, 6.3) |
| | Female | 816 | 141,674 | 5.8 (5.4, 6.2) |
| Age | 15–49 | 407 | 241,559 | 1.7 (1.5, 1.9) |
| | 50–64 | 326 | 20,697 | 15.8 (14.1, 17.5) |
| | 65 + | 892 | 18,206 | 48.9 (45.9,52.2) |

**Table 3. CSMF for the top 20 cause of death using physician review by age group, Butajira HDSS, January 2008- April 2017, Ethiopia.**

| Verbal autopsy title | VA-code | Age category | | | | | | Total (F = 498, M = 490) | |
|---|---|---|---|---|---|---|---|---|---|
| | | 15–49 (F = 108, M = 123) | | 50–64 (F = 118, M = 104) | | 65 and above (F = 272, M = 263) | | | |
| | | N | CSMF(95% CI) | N | CSMF(95% CI) | N | CSMF(95% CI) | N | CSMF(95% CI) |
| Accidental fall | VA-11.03 | 5 | 1.60E-02 | 3 | 1.19E-02 | 9 | 1.50E-02 | 17 | 1.46E-02 |
| | | | (5.21E-03,3.69E-02) | | (2.45E-03,3.43E-02) | | (6.88E-03,2.83E-02) | | (8.52E-03,2.32E-02) |
| Acute lower respiratory infections | VA-01.13 | 12 | 3.83E-02 | 9 | 3.56E-02 | 47 | 7.83E-02 | 68 | 5.83E-02 |
| | | | (2.00E-02,6.60E-02) | | (1.64E-02,6.65E-02) | | (5.81E-02,1.03E-01) | | (4.56E-02,7.34E-02) |
| Asthma | VA-05.02 | 2 | 6.39E-03 | 5 | 1.98E-02 | 27 | 4.50E-02 | 34 | 2.92E-02 |
| | | | (7.75E-04,2.29E-02) | | (6.45E-03, 4.55E-02) | | (2.99E-02,6.48E-02) | | (2.03E-02,4.05E-02) |
| Cerebrovascular diseases | VA-04.03 | 6 | 1.92E-02 | 6 | 2.37E-02 | 24 | 4.00E-02 | 36 | 3.09E-02 |
| | | | (7.07E-03,4.13E-02) | | (8.75E-03, 5.09E-02) | | (2.58E-02,5.89E-02) | | (2.17E-02,4.25E-02) |
| Chronic liver disease | VA-06.02 | 26 | 8.31E-02 | 18 | 7.11E-02 | 25 | 4.17E-02 | 69 | 5.92E-02 |
| | | | (5.50E-02, 1.19E-01) | | (4.27E-02, 1.10E-01) | | (2.71E-02, 6.09E-02) | | (4.63E-02,7.43E-02) |
| Congestive heart failure | VA-04.05 | 5 | 1.60E-02 | 3 | 1.19E-02 | 18 | 3.00E-02 | 26 | 2.23E-02 |
| | | | (5.21E-03,3.69E-02) | | (2.45E-03,3.43E-02) | | (1.79E-02,1.79E-02) | | (1.46E-02,3.25E-02) |
| Diabetes mellitus | VA-03.03 | 6 | 1.92E-02 | 7 | 2.77E-02 | 20 | 3.33E-02 | 33 | 2.83E-02 |
| | | | (7.07E-03,4.13E-02) | | (1.12E-02,5.62E-02) | | (2.05E-02,5.10E-02) | | (1.96E-02,3.95E-02) |
| Epilepsy | VA-08.02 | 13 | 4.15E-02 | 1 | 3.95E-03 | 4 | 6.67E-03 | 18 | 1.54E-02 |
| | | | (2.23E-02,7.00E-02) | | (1.00E-04, 2.18E-02) | | (1.82E-03,1.70E-02) | | (9.17E-03,2.43E-02) |
| Gastric and duodenal ulcer | VA-06.01 | 6 | 1.92E-02 | 10 | 3.95E-02 | 17 | 2.83E-02 | 33 | 2.83E-02 |
| | | | (7.07E-03,4.13E-02) | | (1.91E-02,7.15E-02) | | (1.66E-02,4.50E-02) | | (1.96E-02,3.95E-02) |
| HIV/AIDS | VA-01.09 | 42 | 1.34E-01 | 16 | 6.32E-02 | 3 | 5.00E-03 | 61 | 5.23E-02 |
| | | | (9.84E-02,1.77E-01) | | (3.66E-02,1.01E-01) | | (1.03E-03,1.45E-02) | | (4.03E-02,6.67E-02) |
| Hypertensive diseases | VA-04.01 | 12 | 3.83E-02 | 19 | 7.51E-02 | 46 | 7.67E-02 | 77 | 6.60E-02 |
| | | | (2.00E-02,6.60E-02) | | (4.58E-02,1.15E-01) | | (5.67E-02,1.01E-01) | | (5.25E-02,8.18E-02) |
| Intestinal infectious diseases | VA-01.01 | 8 | 2.56E-02 | 15 | 5.93E-02 | 43 | 7.17E-02 | 66 | 5.66E-02 |
| | | | (1.11E-02,4.97E-02) | | (3.36E-02,9.59E-02) | | (5.23E-02,9.53E-02) | | (4.40E-02,7.15E-02) |
| Ischaemic heart disease | VA-04.02 | 2 | 6.39E-03 | 10 | 3.95E-02 | 26 | 4.33E-02 | 38 | 3.26E-02 |
| | | | (7.75E-04,2.29E-02) | | (1.91E-02,7.15E-02) | | (2.85E-02,6.29E-02) | | (2.32E-02,4.45E-02) |
| Malaria | VA-01.10 | 13 | 4.15E-02 | 4 | 1.58E-02 | 17 | 2.83E-02 | 34 | 2.92E-02 |
| | | | (2.23E-02,7.00E-02) | | (4.32E-03,4.00E-02) | | (1.66E-02,4.50E-02) | | (2.03E-02,4.05E-02) |
| Malignant neoplasms of digestive organs | VA-02.01–06 | 10 | 3.19E-02 | 16 | 6.32E-02 | 31 | 5.17E-02 | 57 | 4.89E-02 |
| | | | (1.54E-02,5.80E-02) | | (3.66E-02,1.01E-01) | | (3.54E-02,7.25E-02) | | (3.72E-02,6.29E-02) |
| Mental disorder, unspecified | VA-08.99 | 8 | 2.56E-02 | 6 | 2.37E-02 | 13 | 2.17E-02 | 27 | 2.32E-02 |
| | | | (1.11E-02,4.97E-02) | | (8.75E-03, 5.09E-02) | | (1.16E-02,3.68E-02) | | (1.53E-02,3.35E-02) |

*(Continued)*

**Table 3.** (Continued)

| Verbal autopsy title | VA-code | Age category | | | | | | | | Total (F = 498, M = 490) | |
|---|---|---|---|---|---|---|---|---|---|---|---|
| | | 15–49 (F = 108, M = 123) | | 50–64 (F = 118, M = 104) | | 65 and above (F = 272, M = 263) | | | | | |
| | | N | CSMF(95% CI) | N | CSMF(95% CI) | N | CSMF(95% CI) | | | N | CSMF(95% CI) |
| Neoplasm of uncertain or unknown behaviour, unspecified | VA-02.99 | 8 | 2.56E-02 (1.11E-02,4.97E-02) | 10 | 3.95E-02 (1.91E-02,7.15E-02) | 25 | 4.17E-02 (2.71E-02, 6.09E-02) | | | 43 | 3.69E-02 (2.68E-02,4.94E-02) |
| Pulmonary tuberculosis | VA-01.03 | 32 | 1.02E-01 (7.10E-02,1.41E-01) | 39 | 1.54E-01 (1.12E-01,2.05E-01) | 87 | 1.45E-01 (1.18E-01, 1.76E-01) | | | 158 | 1.36E-01 (1.16E-01,1.57E-01) |
| Renal failure | VA-07.01 | 12 | 3.83E-02 (2.00E-02,6.60E-02) | 13 | 5.14E-02 (2.76E-02,8.63E-02) | 40 | 6.67E-02 (4.81E-02,8.97E-02) | | | 65 | 5.57E-02 (4.33E-02,7.05E-02) |
| Reproductive neoplasms | VA-02.09–12 | 3 | 9.58E-03 (1.98E-03,2.78E-02) | 12 | 4.74E-02 (2.47E-02,8.14E-02) | 13 | 2.17E-02 (1.16E-02,3.68E-02) | | | 28 | 2.40E-02 (1.60E-02,3.45E-02) |

*F-female, M-male, CSMF-Cause Specific Mortality Fraction

surveillance years, digestive neoplasms were the leading CoD in all ages and responsible for 18.1% of all deaths. Tuberculosis (10.7%) and HIV/AIDS-related deaths (10%) were the second and third leading causes of death, respectively. The specific CoD determined in this surveillance period further showed that digestive neoplasms, road traffic accidents, and HIV/AIDS-

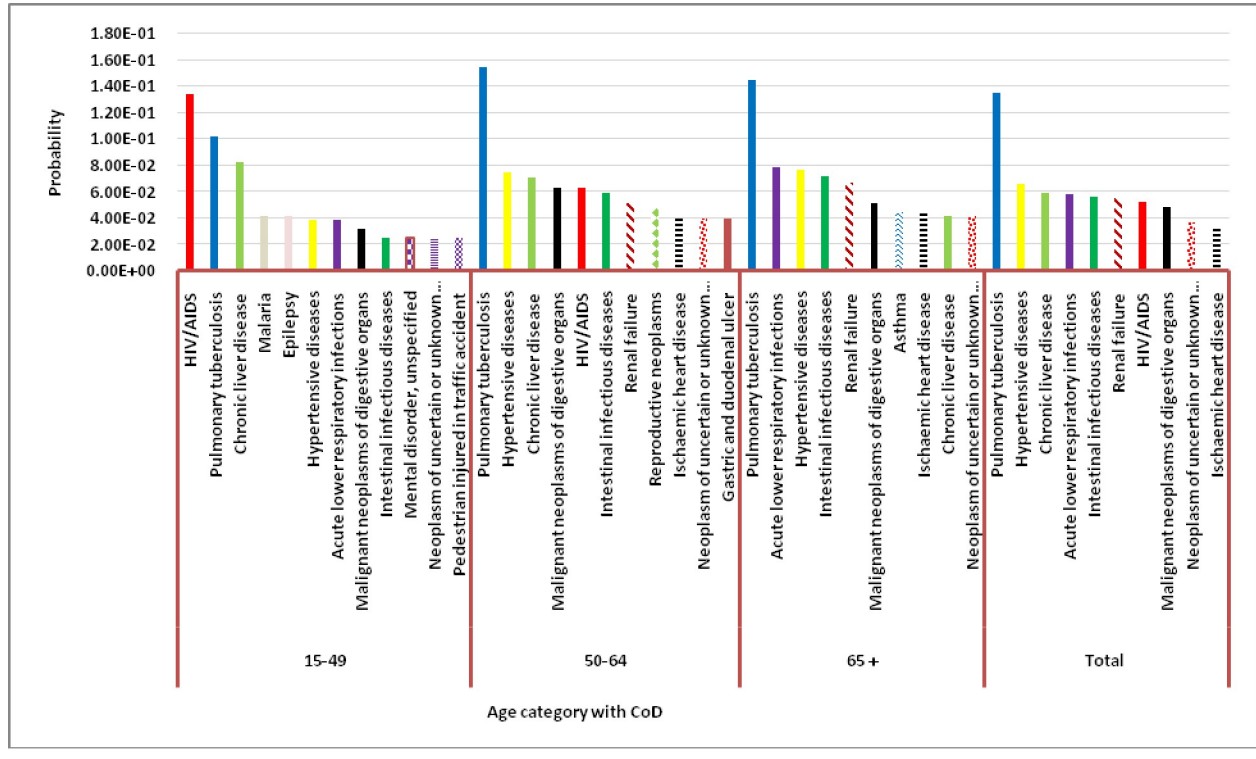

**Fig 1. The probability of dying from ten top specific causes of death across the age category BRHP, January 2008- April 2017.**

**Table 4. Top 20 specific cause of death by age group, Butajira HDSS, May 2017- April 2020, Ethiopia.**

| VA title with WHO VA CoD code | Age category | | | | | | Total (F = 191, M = 180) | |
| | 15–49 (F = 22, M = 41) | | 50–64 (F = 34, M = 28) | | 65 and above (F = 135, M = 111) | | | |
| | N | CSMF (95% CI) | N | CSMF(95% CI) | N | CSMF(95% CI) | N | CSMF(95% CI) |
|---|---|---|---|---|---|---|---|---|
| 06.01 Acute abdomen | 1 | 1.25E-02 | 3 | 4.55E-02 | 8 | 3.09E-02 | 12 | 2.96E-02 |
| | | (3E-04, 6.77E-02) | | (9.47E-03,1.27E-01) | | (1.34E-02, 6.00E-02) | | (1.54E-02,5.12E-02) |
| 04.01 Acute cardiac disease | 2 | 2.50E-02 | 1 | 1.52E-02 | 2 | 7.72E-03 | 5 | 1.23E-02 |
| | | (3.04E-03,8.74E-02) | | (3.84E-04,8.16E-02) | | (9.37E-04,2.76E-02) | | (4.02E-03, 2.86E-02) |
| 01.02 Acute resp infect incl pneumonia | 3 | 3.75E-02 | - | - | 6 | 2.32E-02 | 9 | 2.22E-02 |
| | | (7.80E-03,1.06E-01) | | | | (8.55E-03, 4.97E-02) | | (1.02E-02, 4.18E-02) |
| 02.04 Breast neoplasms | 3 | 3.75E-02 | 3 | 4.55E-02 | 3 | 1.16E-02 | 9 | 2.22E-02 |
| | | (7.80E-03,1.06E-01) | | (9.47E-03,1.27E-01) | | (2.40E-03, 3.35E-02) | | (1.02E-02, 4.18E-02) |
| 03.03 Diabetes mellitus | 2 | 2.50E-02 | 2 | 3.03E-02 | 12 | 4.63E-02 | 16 | 3.95E-02 |
| | | (3.04E-03,8.74E-02) | | (3.69E-03,1.05E-01) | | (2.42E-02,7.95E-02) | | (2.27E-02, 6.34E-02) |
| 01.01 Diarrhoeal diseases | - | - | - | - | 5 | 1.93E-02 | 5 | 1.23E-02 |
| | | | | | | (6.30E-03,4.45E-02) | | (4.02E-03, 2.86E-02) |
| 02.02 Digestive neoplasms | 11 | 1.38E-01 | 17 | 2.58E-01 | 42 | 1.62E-01 | 70 | 1.73E-01 |
| | | (7.07E-02, 2.33E-01) | | (1.58E-01, 3.80E-01) | | (1.19E-01,2.13E-01) | | (1.37E-01, 2.13E-01) |
| 01.03 HIV/AIDS related death | 9 | 1.13E-01 | 8 | 1.21E-01 | 17 | 6.56E-02 | 34 | 8.40E-02 |
| | | (5.28E-02,2.03E-01) | | (5.38E-02,2.25E-01) | | (3.87E-02, 1.03E-01) | | (5.88E-02, 1.15E-01) |
| 06.02 Liver cirrhosis | 1 | 1.25E-02 | 3 | 4.55E-02 | 6 | 2.32E-02 | 10 | 2.47E-02 |
| | | (3E-04, 6.77E-02) | | (9.47E-03,1.27E-01) | | (8.55E-03, 4.97E-02) | | (1.19E-02, 4.49E-02) |
| 04.99 Other and unspecified cardiac dis | - | - | 2 | 3.03E-02 | 4 | 1.54E-02 | 6 | 1.48E-02 |
| | | | | (3.69E-03,1.05E-01) | | (4.22E-03,3.91E-02) | | (5.46E-03,3.20E-02) |
| 01.99 Other and unspecified infect dis | 1 | 1.25E-02 | 1 | 1.52E-02 | 3 | 1.16E-02 | 5 | 1.23E-02 |
| | | (3E-04, 6.77E-02) | | (3.84E-04,8.16E-02) | | (2.40E-03, 3.35E-02) | | (4.02E-03, 2.86E-02) |
| 98 Other and unspecified NCD | - | - | - | - | 5 | 1.93E-02 | 5 | 1.23E-02 |
| | | | | | | (6.30E-03,4.45E-02) | | (4.02E-03, 2.86E-02) |
| 02.99 Other and unspecified neoplasms | 1 | 1.25E-02 | - | - | 15 | 5.79E-02 | 16 | 3.95E-02 |
| | | (3E-04, 6.77E-02) | | | | (3.28E-02, 9.37E-02) | | (2.27E-02, 6.34E-02) |
| 01.09 Tuberculosis | 6 | 7.50E-02 | 8 | 1.21E-01 | 35 | 1.35E-01 | 49 | 1.21E-01 |
| | | (2.80E-02,1.56E-01) | | (5.38E-02,2.25E-01) | | (9.60E-02,1.83E-01) | | (9.09E-02,1.57E-01) |
| 07.01 Renal failure | 3 | 3.75E-02 | 2 | 3.03E-02 | 19 | 7.34E-02 | 24 | 5.93E-02 |
| | | (7.80E-03,1.06E-01) | | (3.69E-03,1.05E-01) | | (4.47E-02,1.12E-01) | | (3.83E-02,8.69E-02) |
| 02.09–12 Reproductive neoplasms MF | 4 | 5.00E-02 | 2 | 3.03E-02 | 17 | 6.56E-02 | 23 | 5.68E-02 |
| | | (1.38E-02,1.23E-01) | | (3.69E-03,1.05E-01) | | (3.87E-02, 1.03E-01) | | (3.63E-02, 8.40E-02) |
| 02.03 Respiratory neoplasms | - | - | 1 | 1.52E-02 | 4 | 1.54E-02 | 5 | 1.23E-02 |
| | | | | (3.84E-04,8.16E-02) | | (4.22E-03,3.91E-02) | | (4.02E-03, 2.86E-02) |
| 12.01 Road traffic accident | 10 | 1.25E-01 | 1 | 1.52E-02 | 1 | 3.86E-03 | 12 | 2.96E-02 |
| | | (6.16E-02,2.18E-01) | | (3.84E-04,8.16E-02) | | (9.77E-05,2.13E-02) | | (1.54E-02,5.12E-02) |
| 03.02 Severe malnutrition | 1 | 1.25E-02 | 2 | 3.03E-02 | 15 | 5.79E-02 | 18 | 4.44E-02 |
| | | (3E-04, 6.77E-02) | | (3.69E-03,1.05E-01) | | (3.28E-02, 9.37E-02) | | (2.66E-02, 6.93E-02) |
| 04.02 Stroke | 5 | 6.25E-02 | 6 | 9.09E-02 | 27 | 1.04E-01 | 38 | 9.38E-02 |
| | | (2.06E-02, 1.40E-01) | | (3.41E-02,1.87E-01) | | (6.98E-02,1.48E-01) | | (6.73E-02,1.27E-01) |

*F-female, M-male, CSMF-Cause Specific Mortality Fraction, CoD = -Cause of Death, VA-Verbal Autopsy

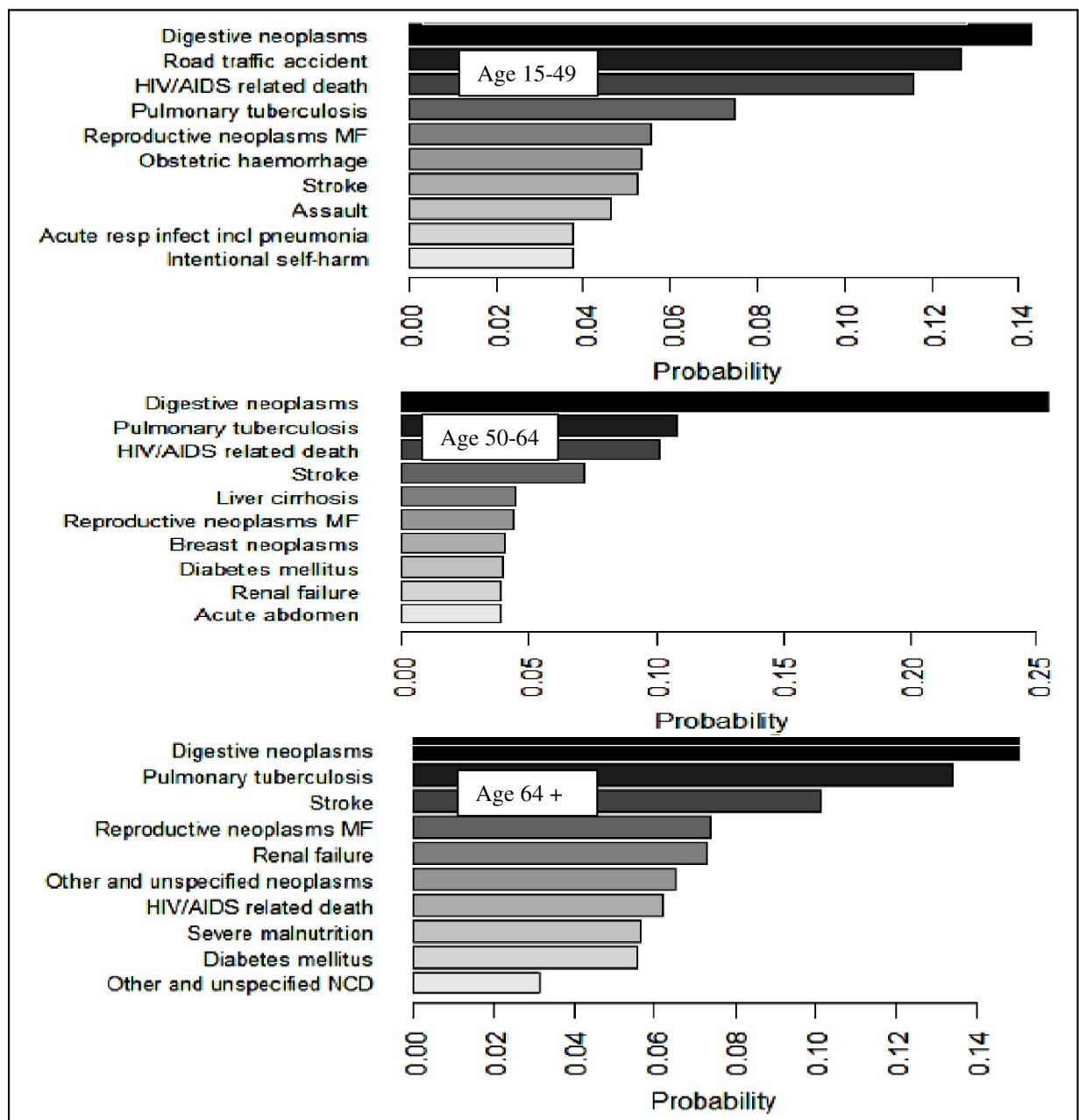

**Fig 2. Top 10 causes of death and their probabilities for adult age categories, Butajira HDSS, May 2017-April 2020, Ethiopia.**

related death were the first (15.2%), second (12.4%), and third (11.3%) leading causes of death in the age group 15–49, respectively. In the age group 50–64, 25.4%, 10.8%, and 10.2% of deaths were attributed to digestive neoplasms, tuberculosis, and HIV/AIDS-related death and were the first, second, and third leading causes of death, respectively. Digestive neoplasms (15.1%), tuberculosis (13.7), and stroke (10.2%) were the first, second, and third leading causes of death in old age (65+), respectively.

Moreover, Fig 2 below demonstrates the ten top causes of death across and within the pre-defined age group. Digestive neoplasms were the leading CoD in all circumstances in May

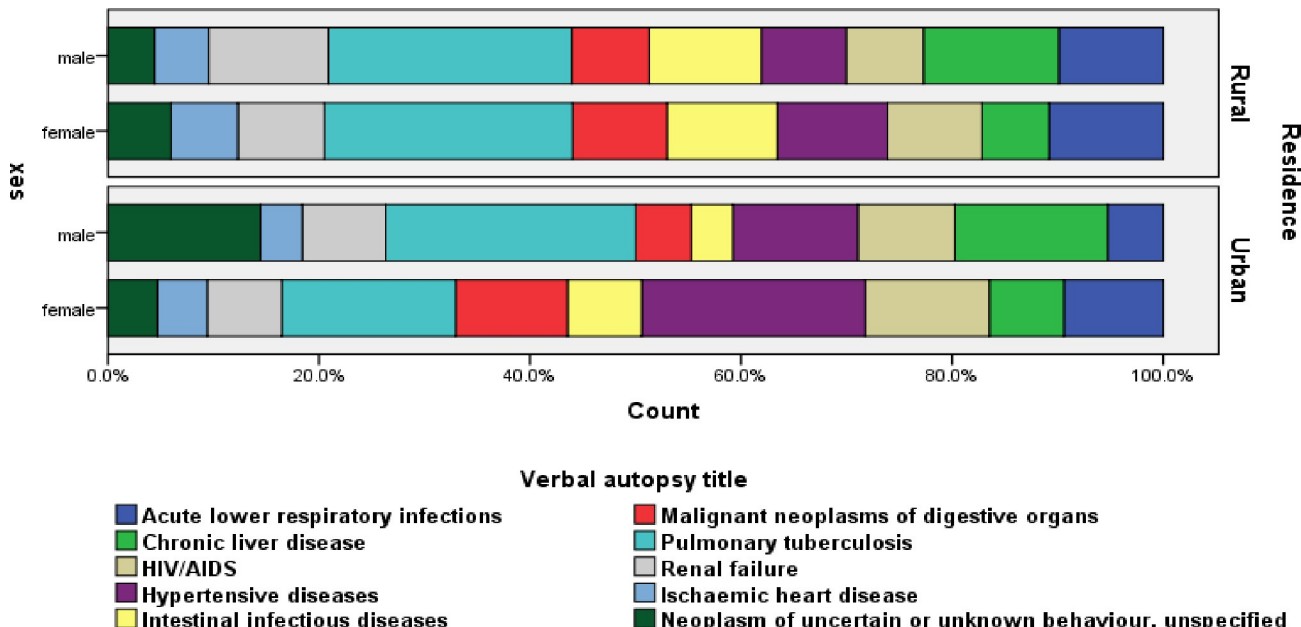

**Fig 3. Top ten specific causes of death using the physician review method by sex and residence, Butajira HDSS, January 2008- April 2017, Ethiopia.**

2017 through April 2020 surveillance period. The CSMF of the age-specific death showed that across the range of causes of death, a randomly selected case had a significantly higher probability of dying from digestive neoplasms (Fig 2).

In the sex-disaggregated analysis of specific causes of death for VA data registered from January 2008 to April 2017, tuberculosis was marginally higher among males (13.8%:13.3%). Still, chronic liver disease was higher in males than females and accounted for 7.8% and 4.0%, respectively. HIV/AIDS and hypertensive diseases were higher in females, accounting for 5.9% and 7.9% than in males, 5.3% and 4.6%, respectively. The proportion of deaths due to renal failure is higher in males (6.3%) than in females (4.8%), whereas malignant neoplasms of digestive organs are higher in females (5.7%) than in males (4.1%) (Fig 3). Similarly, digestive neoplasm and HIV/AIDS-related deaths were higher among females (19.0% and 10.4%) than males (17.4% and 4.6%) from May 2017- April 2020. Tuberculosis and renal failure were higher in males, accounting for 15.9% and 8.2% than in females, accounting for 7.1% and 3.8% of deaths, respectively, and stroke is marginally higher among males (9.7%) than in females (9.0%) (Fig 4).

The rural-urban mortality disparities in the CoD identified during January 2008 to April 2017 surveillance time identified that tuberculosis was higher in rural areas (14.1%) than in urban areas (11.8%). HIV/AIDS and hypertensive diseases were higher in urban areas than rural areas, and responsible for 6.3% and 10.0%, 4.9%, and 5.6% of deaths in urban and rural areas, respectively. Chronic liver disease was marginally lower in rural areas (5.8%) than in urban areas(6.3%). Death due to renal failure is higher in rural areas (5.9%) than in their counterpart urban areas (4.4%) (Fig 3).

Besides, in the next three surveillance years for the period from May 2017 to April 2020, the urban-rural difference in the specific-CoD analysis probed that digestive neoplasm and tuberculosis were higher in urban areas than in rural areas and accounted for 23.1% and 30.8%, 18.1% and 10.7%, respectively. All deaths from HIV/AIDS and renal failure were recorded in rural areas (Fig 4).

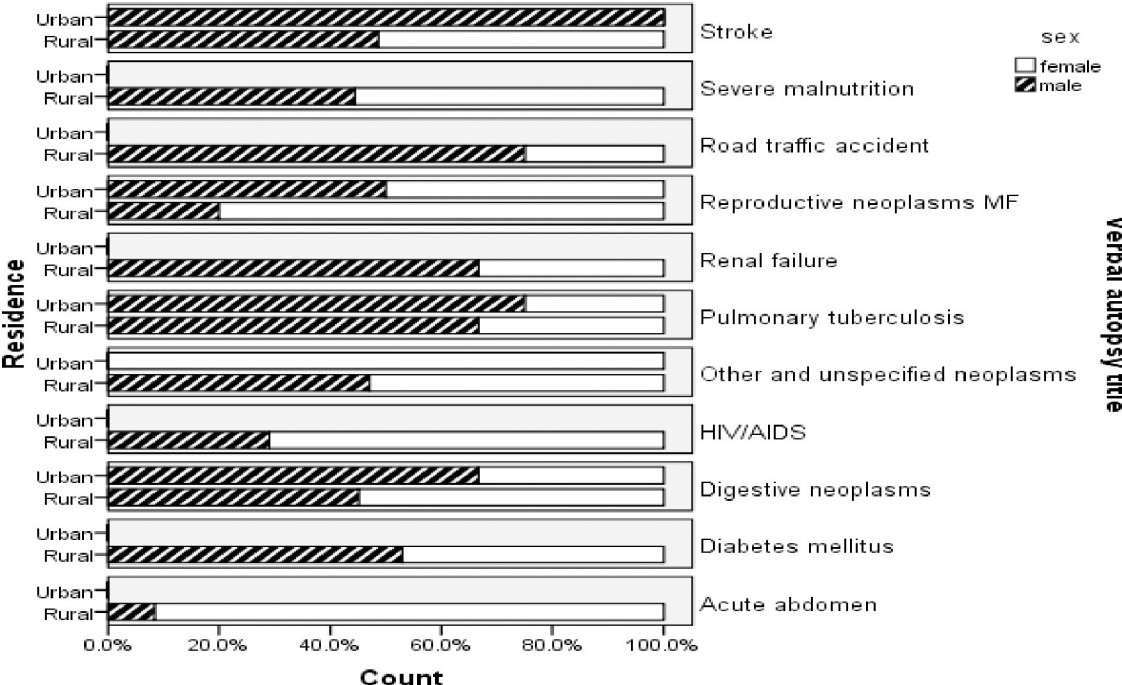

**Fig 4. Top ten specific causes of death using the InterVA model by sex and residence, Butajira HDSS, May 2017- April 2020, Ethiopia.**

## Discussion

In Butajira HDSS, deaths captured through VA surveyed from January 2008 to April 2020 during the first and second surveillance years were interpreted using the physician review and the InterVA methods. The overall adult mortality rate of 5.8 (95%CI: 5.5, 6.0) per 1000py this study identified is lower than the 1987 and 2004 study of 7.8 (95%CI: 7.4,8.0) per 1,000 person-years from Butajira HDSS [31]. This difference may be associated with the age group used in the former Butajira study (15 to 64) and the initiation of the health extension program in the country [32]. The specific CoD primed analysis indicated that during the first follow-up period, tuberculosis was the first leading CoD and the second CoD in the second surveillance year. Similarly, tuberculosis is the leading CoD in several HDSS sites; 15.9% from Kilite Awlaelo [18] and 19.7% from Dabat HDSS [19] in Ethiopia, 26.9% [33] and 16% [34] in Kenya, and 18.5% in South Africa [35]. However, a lower proportion of deaths due to tuberculosis were registered from Ballabgarh in India (7.3%) [36], Dodowa in Ghana (9.7%) [37], Taabo in Cote d'Ivoire (6.4%) [38] and Matlab in Bangladesh (6.1%) [39]. The disparity may be associated with variation in the surveillance years and its duration and HDSS site dissimilarity [40] like disparities in socioeconomic status, exposure risk difference [41], including the way individuals live, work and interact, and the healthcare system capacity in managing tuberculosis cases [42]. Besides, in a previous Butajira HDSS study, 9.7% of deaths were attributed to tuberculosis [43]; this disparity may also be associated with a methodological difference in the CoD determination procedure (computer algorithm) and age category (restricted to the 15–49 age range).

Ethiopia has made significant progress towards achieving the Millennium Development Goals (MDGs) [44] and endorsed the 2030 agenda for Sustainable Development Golas (SDGs) [32]. Even though the Ethiopian government has been striving to avert the burden of

tuberculosis through major strategic initiatives like screening, treatment, and follow-up services at all levels [45], it continues to be a significant public health problem [32]. However, still, recent reports indicate that there are missed tuberculosis investigations [46], notable tuberculosis diagnosis delays [47], poor adult tuberculosis treatment outcomes [48], and the existing gap between policy and practice [49].

Digestive neoplasms were the first leading CoD in the second surveillance period. Lower digestive system neoplasms (5.5–6.0%) in Matlab-HDSS, Bangladesh [50], and 4.5% in Dodowa-HDSS, Ghana [37] were reported. Globally, cancer is responsible for 9.0 million deaths annually [51]. Of cancer-related deaths worldwide, digestive system tumors are the most frequently diagnosed malignancies [50, 52, 53] and are responsible for 35% of all cancer-related deaths [54]. In Ethiopia, the 2020 Global Cancer report showed that of all new cancer cases, colorectal cancer accounted for 7.8% [24]. Based on a population-based cancer registry [55] and a government report in Ethiopia [56], colorectal cancers are the most common type. Low awareness and inadequate screening and treatment services can explain high cancer mortality [45]. Age, hormones, immunosuppression, diet, infectious agents, cancer-causing substances and radiation, alcohol consumption, and tobacco use are usually considered risk factors for cancer development [57]. In LMICs, a significant transition has been observed due to the decline in mortality from major infectious diseases and the growing burden of chronic diseases like cancer [58, 59]. Different stages of an epidemiological transition are happening in different African countries [60]. In this context, the current findings support the previous evidence about the epidemiological transition in the Butajira population [31].

Hypertensive diseases and stroke were the common CoD during the entire surveillance period. Hypertension accounted for 31.9% of Dabat-HDSS [20], whereas 4.4% of Agincourt–HDSS in South Africa [35]. Of 1.28 billion estimated hypertensive cases worldwide, two-thirds live in LMICs [61]. In Ethiopia, hypertension affects 15.9% of the population, only 1.5% was controlled, and 2.8% received appropriate treatment and care [23]. It is expected that up to 80% of strokes and 40% of cancers could be prevented by controlling the exposure risks [32]. The 2015 Ethiopian government report indicated that about 16% of the population is hypertensive [45], and stroke is becoming an alarming public health problem in the country [62, 63]. The 2019 global burden of disease report also indicated that stroke is one cause of premature mortality in Ethiopia [56, 64]. Despitestrategic direction to reduce the burden of hypertension in place, undiagnosed hypertension is still high in Ethiopia [65] with significant risk factors [66]. Besides, the available health services are minimal [45].

Besides, a considerable CoD due to chronic liver disease was observed in the first surveillance time. Similarly, chronic liver disease (8.8%) was the third leading CoD in Kersa HDSS in Ethiopia [67], and this might be explained by the availability and the social habit of chewing indigenous chewing plant 'khat' as a stimulant [68] and other evidence also indicated that a regular khat chewing habit predisposed individuals with impaired liver function [69].

In the first surveillance years, the age-disaggregated analysis indicated that HIV/AIDS was the leading CoD in younger adults. During the second surveillance time, it was a considerable CoD in the age group 15–60. A similar finding was reported in the age group 15–49 in Ethiopia; Kilite Awlaelo HDSS,10.9% [18] and 16.3% [70], and a common CoD from Dabat HDSS [19]. It is more common among adults aged 15–49 years [71, 72], associated with developmental, psychological, biological, social, and other factors [73]. WHO recently estimated that young individuals (15–25 years) contributed to over 30% of all new HIV infections globally [74]. However, Ethiopia has one of the lowest HIV prevalence rates [30]. The disparity may be associated with variations in the method of case determination. The Ethiopian government has set up a strategic direction to end HIV/AIDS as a public health threat by 2030 [45, 75]. Besides, Ethiopia adopted and implemented the global 90–90–90 HIV prevention [75]. A

lower prevalence of HIV was reported in the 2011 Ethiopian Demographic and Health Survey. However, there are various challenges regarding primary HIV prevention [22], and recent reports indicated the resurgence of HIV/AIDS in Ethiopia among young adults [75] and urged a sustainable control strategy [76].

In the age 50+, tuberculosis was the most common CoD across the entire surveillance years. Similarly, it is the leading CoD in the age group 50+ from Kilite Awlaelo-[18, 70] and Dabat-HDSS [19], Ethiopia, in the age group (15+) from Dande-HDSS, Angola [77], in the age group (15–64) from Matlab-HDSS, Bangladesh [50] and in the age group (15–49) from Pur-worejo-HDSS, Indonesia [78]. Variations in the surveillance period [40] and exposure risk differences in tuberculosis infections [41] may explain the differences. The 2020 WHO report indicated that tuberculosis is causing one-third of AIDS-related deaths globally [79]. In Ethiopia, TB/HIV collaborative activities, including initiating and scaling up free ART services, helped to reduce tuberculosis and HIV mortality [32]. Moreover, the age-stratified analysis using the model indicated that digestive neoplasms were the leading specific CoD in all adult age groups. In Ethiopia, the risk of dying from cancer is 9·4% in individuals under the age of 75 years [56]. Although digestive neoplasms are frequently reported in older adults (over 50), they are becoming increasingly common in younger adults worldwide [80]. During the first surveillance time, hypertensive diseases were the common CoD in the age group 50+ and stroke in the old age (65+) in the second surveillance year. Similarly, hypertensive disease was common in the age group 50+ in Dabat HDSS, Ethiopia [19]. Additionally, in the Arba Minch HDSS, Ethiopia, hypertension frequently impacted the elder age group [81], and stroke was the leading CoD among older adults in Indonesia [78].

A significant proportion of deaths due to traffic accidents were identified throughout the surveillance, especially in younger adults. Similarly, road traffic accident deaths are commonly reported in six HDSS sites in Ethiopia [82], and men of younger adults are more exposed to road injuries [82, 83]. Annually, road traffic accident kills approximately 1.3 million people globally and 90% occur in LMICs [84]. Due to insufficient efforts (like capacity and resource limitations) and policy gaps to ensure road safety [83], road traffic accidents in Ethiopia are increasing at an alarming rate [85]. Inadequate traffic law enforcement, unsafe vehicles and road infrastructure, distracted and influenced driving, and speeding are among the most common causes of road traffic crashes [84]. However, the United Nations planned to prevent half of the deaths from road traffic accidents by 2030 [84].

Moreover, tuberculosis and chronic liver disease were higher in males, but digestive neoplasms, HIV/AIDS and hypertensive diseases were higher in females. Likewise, males had a 46% lower risk of death due to HIV/AIDS than females in Dabat HDSS, Ethiopia [19]. Adolescent girls and young women in sub-Saharan Africa are at the highest risk of contracting HIV [71]. In a VA-based study in Kenya, tuberculosis was higher in males (16.6%) than in females (15.8%), and HIV/AIDS was higher in females (35.5%) than males (27.5%) [34]. Deaths due to tuberculosis were higher in males in Ballabgarh HDSS, India [36, 39]. WHO recently reported that 56% and 32% of tuberculosis-developed cases were attributed to men and women (aged ≥15 years), respectively [86]. However, in women aged 50+, tuberculosis was the major CoD in Dabat-HDSS, Ethiopia [19]. Moreover, in Ifakara-HDSS Tanzania, tuberculosis was higher in women 30–44 years old, and HIV/AIDS and tuberculosis were the second and third CoD in 45+ age groups for both sexes [87]. Furthermore, digestive neoplasm deaths were higher among females. However, the Global burden of cancer study in Ethiopia showed that colorectal cancer accounted for 10.6% of deaths in males than 9.4% in females [24]. Digestive neoplasms were common in men in Matlab-HDSS, Bangladesh [50] and Farafenni HDSS, Gambian [88]. But, still, digestive neoplasms are reported as prominent in both sexes [38].

The rural-urban mortality disparities in the CoD identified tuberculosis as higher in rural areas, whereas HIV/AIDS and hypertensive diseases were more common in urban areas. Similarly, higher HIV/AIDS deaths in urban were reported in Butajira-[43] and Dabat-HDSS [19]. Because of inadequate basic life and health need, the risk of death from tuberculosis was higher among rural residents [67]. In Ethiopia, a higher prevalence of hypertension was reported in urban areas (22%) than in rural (13%) [45]. Hypertension is also linked to urbanization, where people's lifestyles change [70].

## Strengths and limitations of the study

Providing well-designed VA-based CoD evidence with a wide spectrum of applications for policymakers and relevant stakeholders, where registration of deaths is desperately available in the local community, would be considered the main strength of this study. However, this study suffered from two different VA interpretation techniques. The reliability and validity of VA CoD assigning and interpretation processes, particularly for diseases with less distinct clinical pictures, are subject to several constraints. The inter-observer variation in the physician review method and the rigidity of the computerized InterVA algorithm in identifying CoD can impair the VA's accuracy compared to the actual gold standard for the CoD diagnosis. Although the repetition of personal ID and other related issues have tried to be solved, a small percentage of cases with inconsistent data were excluded from the current analysis.

## Conclusion

Key findings from this study highlight that tuberculosis was the leading CoD during the 2008–2017 follow-up periods. In the second phase of the surveillance time (2017–2020), digestive neoplasms were the leading CoD. Hypertensive diseases and HIV/AIDS-related deaths were also common causes of death in the Butajira community. When deaths were disaggregated by socio-demographic characteristics of the deceased, tuberculosis mortality was higher in persons aged 50+, among males and in rural areas during 2008-2017. On the other hand, in the second surveillance period, digestive neoplasms were higher in the age group (50–64) and higher in females. Generally, tuberculosis was the leading CoD in the earlier surveillance years. The significant increase in digestive neoplasms associated with mortality in the later surveillance period is the probable indication of epidemiological transition in the community. Hence, early detection and prompt treatment of tuberculosis cases should be emphasized to achieve an end tuberculosis strategy, primarily focusing on the transmission dynamics-oriented feasible prevention practices. Besides, prevention, early detection with appropriate follow-up, and management of cases at all levels of the existing healthcare system must be prioritized to avert premature deaths due to digestive neoplasms. On top of that, initiating and strengthening a comprehensive cancer surveillance system, including a population-based cancer registry, is an essential add-on to tackle the emerging problem.

## Supporting information

**S1 Text. Disease classifications and corresponding verbal autopsy codes.**
(DOC)

## Acknowledgments

We would like to acknowledge the BRHP management committee, leaders of the program, and the Butajra Community members. We would like to extend our sincere gratitude to Dr. Wubetsh Asnake for her professional support. We are very grateful to Tesfamichael Awoke

(director of the Butajra HDSS) and the team: Etsehiwot Tilahun, Genamo Irensa and Lemma Gonfa. We also appreciate VA field team leaders Abiot Weldemariam and Mulugeta Tadesse, and database managers Kasahun Shiferaw and Gemechu Getachow.

## Author Contributions

**Conceptualization:** Hailelule Aleme, Wubegzier Mekonnen, Alemayehu Worku.

**Formal analysis:** Hailelule Aleme.

**Investigation:** Hailelule Aleme.

**Methodology:** Hailelule Aleme, Wubegzier Mekonnen, Alemayehu Worku.

**Software:** Hailelule Aleme.

**Visualization:** Hailelule Aleme.

**Writing – original draft:** Hailelule Aleme, Alemayehu Worku.

**Writing – review & editing:** Wubegzier Mekonnen.

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
