## [Decision Letter · Decision Letter 0]

6 Jun 2022

PGPH-D-21-01036

Cause-Specific Mortality Fraction (CSMF) of Adult Mortality in Butajira, South Central Ethiopia

Dear Dr. Yizengaw,

Thank you for submitting your manuscript to PLOS Global Public Health. After careful consideration, we feel that it has merit but does not fully meet PLOS Global Public Health’s publication criteria as it currently stands. Therefore, we invite you to submit a revised version of the manuscript that addresses the points raised during the review process.

Please submit your revised manuscript by . If you will need more time than this to complete your revisions, please reply to this message or contact the journal office at globalpubhealth@plos.org. Please include the following items when submitting your revised manuscript:

We look forward to receiving your revised manuscript.

Kind regards,

Kashif Shafique

Academic Editor

Journal Requirements:

2. Please note that your Data Availability Statement is currently missing [the repository name and/or the DOI/accession number of each dataset OR a direct link to access each database]. If your manuscript is accepted for publication, you will be asked to provide these details on a very short timeline. We therefore suggest that you provide this information now, though we will not hold up the peer review process if you are unable.

Additional Editor Comments (if provided):

Reviewers' comments:

Reviewer's Responses to Questions

**Comments to the Author**

1. Does this manuscript meet PLOS Global Public Health’s publication criteria? Is the manuscript technically sound, and do the data support the conclusions? The manuscript must describe methodologically and ethically rigorous research with conclusions that are appropriately drawn based on the data presented.

Reviewer #1: Partly

Reviewer #2: Partly

2. Has the statistical analysis been performed appropriately and rigorously?

Reviewer #1: No

Reviewer #2: No

3. Have the authors made all data underlying the findings in their manuscript fully available (please refer to the Data Availability Statement at the start of the manuscript PDF file)?

Reviewer #1: Yes

Reviewer #2: Yes

4. Is the manuscript presented in an intelligible fashion and written in standard English?

Reviewer #1: Yes

Reviewer #2: No

5. Review Comments to the Author

Reviewer #1: Major concerns:

In introduction authors build up the justification of using CSMFs without commenting on why authors choose to do analysis using CSMFs instead of cause-specific mortality rates.

Authors have completely ignored to use any measures of validity for both VA methods i.e. PCVA and CCVA. Furthermore, did authors calculated average relative error (ARE) for the estimated CSMFs? If not, please do calculate and report them for all cause of death.

It should be clearly stated whether there were any modification and/ or changes occurred in the Verbal Autopsy questionnaires and its methodology during the study period from 2008 to 2020.

Authors report the use of final likelihood of at least 0.4. Kindly describe the scientific rationale behind selection of this likelihood.

For model specification, authors predefined the prevalence of HIV/AIDS & malaria to be “low”. Citing reference # 29 & 30. Contradicting this, 11th para in discussion stated higher prevalence of HIV. Please elaborate what is the true situation of the disease and use latest statistics especially in a scenario where such statistics are used to define model parameters.

Method section stated that residual intermediate fraction was also calculated in addition to CoD. Whereas, in result section none of the tables show any un-classified cause of death or residual intermediate cause of death.

To further make sense of the reported CSMF, I would highly recommend to present these results along with 95% confidence intervals.

Minor concerns:

Please provide citation for “In many developing nations, cause of death data in a sample of the population can be estimated using CSMFs with a low average error.”

In 3rd para of introduction, authors reported “In Ethiopia, electronic VA databases …”. Whereas, in method sections, it is stated that computer-coded verbal autopsy model (i.e. InterVA) was used in second phase between May 2017 – April 2020. Kindly rectify this statement.

Authors stated, “It is evidenced that there is an epidemiologic transition in the least-income countries, including Ethiopia”. When authors say “it is evidenced …” then please flourish your statement with references.

Please use consistent abbreviations for Verbal Autopsy codes throughout the manuscript (example VAs-01 vs Vas-01-9).

Authors should explicitly mention that this a secondary analysis and should direct readers with appropriate references for further description on the methodology of primary data.

In methods, section for “Disease classification” is un-necessarily descriptive in nature. This information can be presented in a tabular format with few examples from VA with their corresponding ICD-10 classification.

Reviewer #2: Title: Cause-Specific Mortality Fraction (CSMF) of Adult Mortality in Butajira, South Central Ethiopia

PGPH-D-21-01036

Journal: PLOS Global Public Health

The current study is a descriptive analysis of the Cause-Specific Mortality Fraction (CSMF) of adult mortality in Butajira, South Central Ethiopia using two different methods. The objective seems sound, but the introduction and discussion require tightening to more clearly situate the rationale and findings within related literature, as well as to consider the implications of findings for programming & policy. I would suggest using latest and updated published literature instead old fact sheets and reports.

The manuscript suffers from poor quality writing. It needs careful editing for grammar, usage, and punctuation. A proof read by a native English speaker to improve quality is required.

Introduction needs to provide references with the current rate and figures of CSM in Ethiopia.

In first paragraph, reported literature is not matching with cited reference.

Last paragraph of introduction need reference.

First paragraph of Method section should be concise with study settings. Similarly, second paragraph which explained disease classification is additional; this is not a methodology part.

The authors used two methods for assigning the cause of death. Why did you use both methods? Which method was more appropriate to explain the specific cause? Just showing the results were too much descriptive.

The methods section does not provide details for the computing software/package of model InterVA.

Authors should provide adult mortality rates by age and sex with addition of Table 1.

I recommend shortened Table 2 and Table 3 by reporting only top 20 specific causes of death. Similarly, Figure 1 and Supplementary table 1 is not informative since it contains the same information that was presented in tables.

Discussion section is not technically prepared. Need to be format. Results and findings are repeated as previously reported in results section.

I think the discussion could be shortened significantly to (1) quickly summarize findings and (2) discuss the implications, including how it relates to other literature in Ethiopia or similar contexts.

Some references are irrelevant.

Need to be updating and should be in a single reference style, journal required format.

6. PLOS authors have the option to publish the peer review history of their article (what does this mean?). If published, this will include your full peer review and any attached files.

**Do you want your identity to be public for this peer review?** For information about this choice, including consent withdrawal, please see our Privacy Policy.

Reviewer #1: **Yes: **Owais Raza

Reviewer #2: No

---

## [Decision Letter · Decision Letter 1]

21 Nov 2022

PGPH-D-21-01036R1

Cause-Specific Mortality Fraction (CSMF) of Adult Mortality in Butajira, South Central Ethiopia

Dear Dr. Yizengaw,

Thank you for submitting your manuscript to PLOS Global Public Health. After careful consideration, we feel that it has merit but does not fully meet PLOS Global Public Health’s publication criteria as it currently stands. Therefore, we invite you to submit a revised version of the manuscript that addresses the points raised during the review process.

We look forward to receiving your revised manuscript.

Kind regards,

Kashif Shafique

Academic Editor

Journal Requirements:

Additional Editor Comments (if provided):

Reviewers' comments:

Reviewer's Responses to Questions

**Comments to the Author**

1. If the authors have adequately addressed your comments raised in a previous round of review and you feel that this manuscript is now acceptable for publication, you may indicate that here to bypass the “Comments to the Author” section, enter your conflict of interest statement in the “Confidential to Editor” section, and submit your "Accept" recommendation.

Reviewer #1: (No Response)

Reviewer #2: All comments have been addressed

2. Does this manuscript meet PLOS Global Public Health’s publication criteria? Is the manuscript technically sound, and do the data support the conclusions? The manuscript must describe methodologically and ethically rigorous research with conclusions that are appropriately drawn based on the data presented.

Reviewer #1: Partly

Reviewer #2: Yes

3. Has the statistical analysis been performed appropriately and rigorously?

Reviewer #1: Yes

Reviewer #2: Yes

4. Have the authors made all data underlying the findings in their manuscript fully available (please refer to the Data Availability Statement at the start of the manuscript PDF file)?

Reviewer #1: No

Reviewer #2: Yes

5. Is the manuscript presented in an intelligible fashion and written in standard English?

Reviewer #1: Yes

Reviewer #2: Yes

6. Review Comments to the Author

Reviewer #1: I would take this opportunity to appraise authors that they have extensively revised this manuscript. But there are some minor issues which authors are still needed to address before this manuscript becomes acceptable for the publication.

1. Authors have clarified why results from two systems cannot be validated due to unavailability of gold standard. I would suggest to produce kappa statistics between two methods to assess whether these two systems are producing similar results.

2. Regarding the final likelihood of 0.4, authors claimed in response report that they have described in revised manuscript. But looking at the revised manuscript, I cannot find their explanation. Instead authors have redirected the readers by stating, "The details were discussed elsewhere[28]". I do not agree that this can be categorized as "scientific rationale".

3. In table # 3, total numbers of deaths for female & male in the last column do not add up with the column total. Kindly rectify this. Similarly, total numbers of deaths for female & male do not match with column totals (age-wise) as well as grand total. Need careful revision in this regards.

Reviewer #2: The authors have responded well and improved the new version of the manuscript based on the comments of the reviewers. I have no further comments.

7. PLOS authors have the option to publish the peer review history of their article (what does this mean?). If published, this will include your full peer review and any attached files.

**Do you want your identity to be public for this peer review?** For information about this choice, including consent withdrawal, please see our Privacy Policy.

Reviewer #1: **Yes: **Owais Raza

Reviewer #2: **Yes: **Sidra Zaheer

---

## [Decision Letter · Decision Letter 2]

17 Jan 2023

Cause-Specific Mortality Fraction (CSMF) of Adult Mortality in Butajira, South Central Ethiopia

PGPH-D-21-01036R2

Dear Mr Yizengaw,

We are pleased to inform you that your manuscript 'Cause-Specific Mortality Fraction (CSMF) of Adult Mortality in Butajira, South Central Ethiopia' has been provisionally accepted for publication in PLOS Global Public Health.

Best regards,

Abraham D. Flaxman, Ph.D.

Academic Editor

Reviewer Comments (if any, and for reference):

Reviewer's Responses to Questions

**Comments to the Author**

1. If the authors have adequately addressed your comments raised in a previous round of review and you feel that this manuscript is now acceptable for publication, you may indicate that here to bypass the “Comments to the Author” section, enter your conflict of interest statement in the “Confidential to Editor” section, and submit your "Accept" recommendation.

Reviewer #1: All comments have been addressed

Reviewer #2: All comments have been addressed

2. Does this manuscript meet PLOS Global Public Health’s publication criteria? Is the manuscript technically sound, and do the data support the conclusions? The manuscript must describe methodologically and ethically rigorous research with conclusions that are appropriately drawn based on the data presented.

Reviewer #1: Yes

Reviewer #2: Yes

3. Has the statistical analysis been performed appropriately and rigorously?

Reviewer #1: Yes

Reviewer #2: Yes

4. Have the authors made all data underlying the findings in their manuscript fully available (please refer to the Data Availability Statement at the start of the manuscript PDF file)?

Reviewer #1: Yes

Reviewer #2: Yes

5. Is the manuscript presented in an intelligible fashion and written in standard English?

Reviewer #1: Yes

Reviewer #2: Yes

6. Review Comments to the Author

Reviewer #1: The authors did a great job enhancing the quality of this work, and based on my review of the final version, I definitely recommend accepting it for publication.

Reviewer #2: I have no further comments.

7. PLOS authors have the option to publish the peer review history of their article (what does this mean?). If published, this will include your full peer review and any attached files.

**Do you want your identity to be public for this peer review?** For information about this choice, including consent withdrawal, please see our Privacy Policy.

Reviewer #1: **Yes: **Owais Raza

Reviewer #2: **Yes: **Sidra Zaheer
